# Can Land Consolidation Reduce the Soil Erosion of Agricultural Land in Hilly Areas? Evidence from Lishui District, Nanjing City

**Yanyuan Zhang [1],[†], Cong Xu [2],[†] and Min Xia [3],***

[1] College of Economics and Management, Nanjing Forestry University, Nanjing 210037, China; yyzhang@njfu.edu.cn
[2] College of Landscape Architecture, Nanjing Forestry University, Nanjing 210037, China; nfuxo@njfu.edu.cn
[3] College of Land Management, Nanjing Agricultural University, Nanjing 210095, China
* Correspondence: xm@njau.edu.cn
[†] These authors contributed equally to this work.

**Abstract:** The hilly areas of China have experienced soil erosion and are also typical land consolidation (LC) regions. Using the RUSLE model and the multiple regression model, this study evaluated the soil erosion of agricultural land and assessed the effects of LC on soil erosion in Lishui District, a typical district in the Ning-Zhen-Yang hilly area. The soil erosion of agricultural land ranged from 0 to 385.77 t·ha$^{-1}$·yr$^{-1}$ with spatial heterogeneity due to the topography, land cover, and vegetation cover. Overall, carrying out LC reduced soil erosion due to the construction of protection forests, farmland shelterbelts, and different kinds of land engineering. Furthermore, the different types of LC had different impacts on soil erosion, where farmland consolidation resulted in more serious soil erosion than land development. Nevertheless, the potential risks brought by LC to soil erosion reduction could not be overlooked, and more attention should be paid to ecological environment protection during the process of LC. This study presents findings regarding the positive impacts and potential risks of LC for soil erosion reduction in agricultural land in hilly areas.

**Keywords:** agricultural land; land consolidation; soil erosion; heterogeneity; RUSLE; multiple regression model

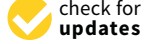



## 1. Introduction

Soil erosion is a global ecological issue that threatens ecosystem health and sustainable development. It is an important cause of reservoir sedimentation and soil nutrient loss, thus it has a significant impact on the eutrophication of water bodies and freshwater pollution, and is also a major threat to soil quality and agricultural productivity [1–6]. Soil erosion is also a common ecological problem in China; according to the Ministry of Water Resources of the Peoples' Republic of China, there was a total soil erosion area of 2.71 million square kilometers in 2019, accounting for 28.34% of the total area. Although soil erosion in hilly areas is not the most serious issue in China, the harm of soil erosion needs to be considered, as more land in these areas is involved in agricultural production, and the lower reaches of rivers are often important industrial and agricultural production bases and economic centers. Therefore, soil erosion and its control in hilly areas deserve more attention and study.

The first issue raised in the study of soil erosion is how to quantitatively estimate the intensity of soil erosion and its spatial distribution. An accurately quantitative assessment of soil erosion dynamics is crucial to understand the erosion process and contributes to soil and water conservation [7,8]. Among the existing models, the universal soil loss equation (USLE) [9] and the revised universal soil loss equation (RUSLE) [10] models are highly recognized and widely used [3,11–14]. It has been demonstrated that the soil erosion in the

Lower Yangtze Basin ranged from 120 to 260 t·ha$^{-1}$·yr$^{-1}$ from 2001 to 2014 based on the RUSLE method [15]. However, using the data from 2017, the soil erosion modulus ranged from 0 to 50 t·ha$^{-1}$·yr$^{-1}$ in most of the areas in Jiangsu Province [16]. The results differed in terms of temporal and spatial heterogeneity. Indeed, soil erosion in a specific region needs to be estimated to establish ecological restoration measures.

How to reduce soil erosion is another issue that needs to be studied. A number of engineering projects can effectively alleviate the soil erosion of agricultural land, among which LC is worthy of attention but has been less studied. Large-scale LC has been carried out since the mid-1990s and has been subject to a national plan since 2008 in China [17]. LC is carried out to reduce the fragmentation of land, to increase the quantity and quality of cultivated land, and to improve infrastructures and farming conditions in fields, thus enhancing the utilization efficiency of land, water, labor, machinery, and other production factors [18–21]. Land use structure, landscape patterns, vegetation coverage, soil properties, as well as ecological functions are inevitably changed during the process of LC [22–27]. Subsequently, either positive or negative impacts are exerted on soil erosion. On the other hand, the connotation of LC has been constantly expanding [14], and eco-environmental protection has gradually become one of the goals of LC [28–30]. Since the concept of the life community of mountains, rivers, forests, fields, lakes, and grasses was first proposed, the Green Development Concept has gone deep into all aspects of social development. As an agricultural engineering measure, LC should pay more attention to ecological protection, and it is necessary to analyze the impact of LC on soil erosion.

The hilly areas in China are typical LC regions, and greater importance needs to be attached to both soil erosion and its control in these areas. The purposes of this study were to: (1) Quantitatively assess the soil erosion in the study area; (2) quantify the impact and heterogeneity of LC on soil erosion; (3) discuss the positive effects and potential risks brought by LC to soil erosion reduction. These results are valuable for understanding how to reduce soil erosion under LC and to provide strategies for LC in the future.

## 2. Materials and Methods

### 2.1. Study Area

The Ning-Zhen-Yang hilly area is the main area of soil erosion in Jiangsu Province. This study took Lishui District, Nanjing City, a typical area in the Ning-Zhen-Yang hilly area, as the study area. Lishui District is located in the southwest of Jiangsu Province, and low mountains and hills account for 72.5% of the total area. The district is located between 118°51′ and 119°14′ E longitude and 31°23′ and 31°48′ N latitude, and has an area of 1067 km$^2$ (Figure 1). The landscape of Lishui District is characterized by plains, low mountains, hilly topography, and rivers, and lakes. The district is composed of five sub-districts and three towns, with a total of 473.3 thousand permanent residents, and the gross domestic product (GDP) was 78.91 billion with a growth of 8.1% in 2018. From 2006 to 2018, the urbanization rate of the Lishui District increased from 45.1% to 61.0%. More than 500 LC projects were completed in this period, which provided adequate land supply for economic growth and urbanization development. It also helped to optimize construction layout and promote agricultural scale management.

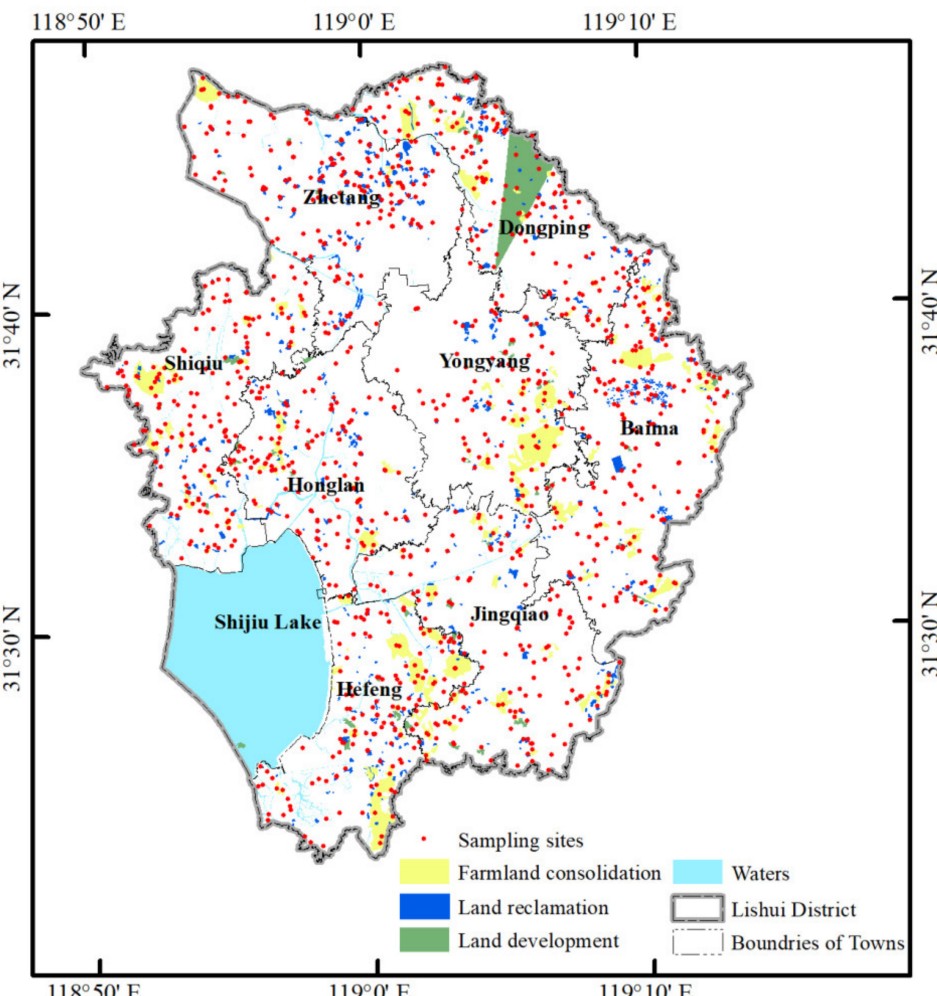

**Figure 1.** Study area and the sampling sites.

*2.2. Data Sources*

The land use data used in this study were extracted from a 1:5000 scale land use map of 2018. The land use types in the study area include farmland (40.76%), forestland (15.36%), grassland (1.36%), water (23.88%), construction land (18.46%), and unused land (0.17%) (Figure 2a). Remote sensing images from Landsat-8 over the same period were collected and used to calculate the normalized difference vegetation index (NDVI). The NDVI is shown in Figure 2b. The ASTER GDEM data (30 m resolution) were obtained from the Geospatial Data Cloud [31]. According to the DEM data, the elevation in this area ranges from 0 to 364 m, and the slope ranges from 0° to 57.19°. The rainfall data were derived from the China Meteorological Data Service Center [32]. The paper soil map of Lishui District was scanned into a digital image. Georeferencing was performed in ArcGIS to integrate the coordinate system with other data. The data of soil types were extracted by vectorization in ArcGIS. The detailed data of the LC projects conducted in Lishui District were collected from the Jiangsu Land Consolidation Center.

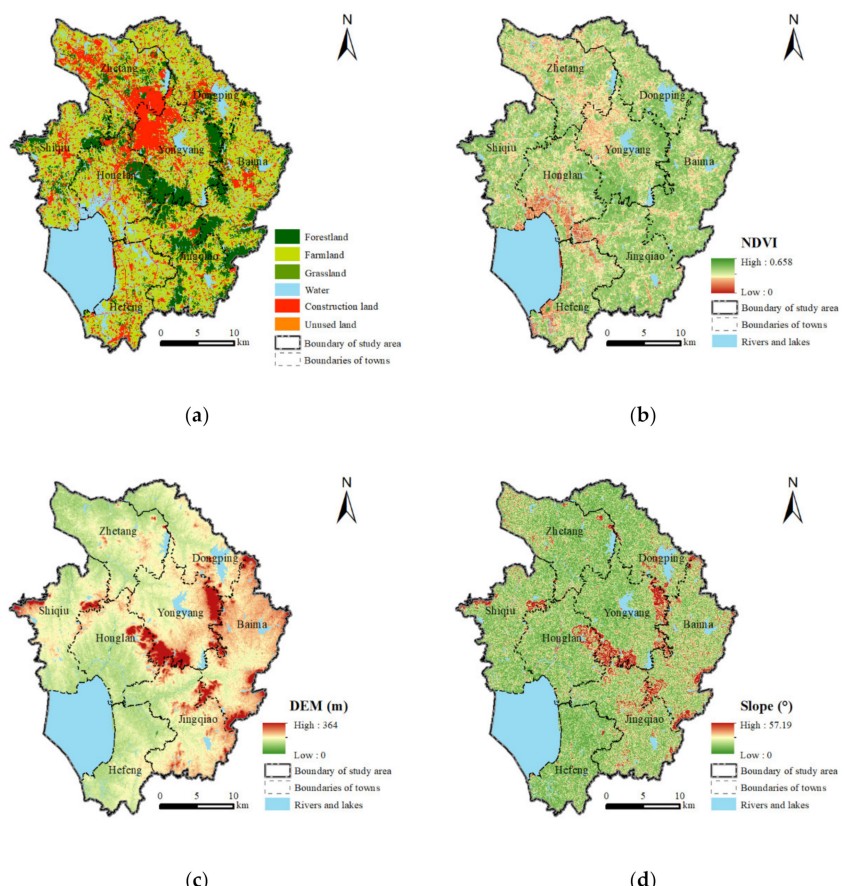

**Figure 2.** (**a**) Land use types and their distribution in the study area; (**b**) NDVI of the study area; (**c**) DEM of the study area; and (**d**) slope of the study area.

For the analysis of the relationship between LC and soil erosion, 300 random points of agricultural land in the LC project area and 300 random points of agricultural land in the non-LC project area were selected using ArcGIS 10.3 (Figure 1). The sampling processes were duplicated for the robustness of the results. The data of dependent variables, independent variables, and control variables were extracted from the collected layers and estimated grids by the sampling sites.

*2.3. The Technique for Calculating Soil Erosion*

In this study, the RUSLE model was used to evaluate the average soil erosion [8,33,34], which can be calculated as a product of five factors and expressed as Equation (1):

$$A = R \times K \times LS \times C \times P \tag{1}$$

where *A* denotes the annual soil losses (t·ha$^{-1}$·yr$^{-1}$), *R* refers to the rainfall erosivity factor (MJ·mm·ha$^{-1}$·h$^{-1}$·yr$^{-1}$), *K* refers to the soil erodibility factor (Mg·ha·h·MJ$^{-1}$·ha$^{-1}$·mm$^{-1}$), LS is the topography factor expressed by slope length and slope steepness (dimensionless), *C* represents the cover management factor (dimensionless), and *P* represents the support practice factor (dimensionless).

The rainfall erosivity factor represents the capacity for soil erosion due to rainfall [9]. The greater the intensity and duration of rainfall, the higher the erosion potential. In this study, the *R* factor was calculated according to Zhang et al. (2002) [35]. This method uses daily rainfall amounts to estimate rainfall erosivity, and has been widely applied in different studies in China [8,33,34].

The soil erodibility factor is a function of soil properties, reflecting the degree of difficulty for soil to be detached and transported [10]. The more difficult it is to erode soil,

the greater the *K* value is, and vice versa. According to the guidelines for the measurement and estimation of soil erosion in production and construction projects, the *K* value in Lishui District is 0.0035 [36].

The length–slope factor reflects the impact of geomorphology on soil erosion. The steeper and longer the slope of a field is, the higher the risk for erosion may be. Due to the data used in the RUSLE included slopes up to 18% [37], the revised formula included slopes greater than 18% [38] is combined to calculate the LS factor [33].

The cover management factor reflects the degree of protection for soil provided by different types of vegetation cover. The potential for soil erosion increases if there is no or very little vegetation cover on the soil. In this study, the *C* factor was calculated according to Cai et al. (2000) [39], which is widely used in studies of China [33,34,38].

As the most difficult factor to determine and the least reliable factor of the RUSLE [10], the support practice factor reflects the impact of soil and water conservation practices. In this study, the *P* factor was obtained from previous studies according to different land use types (Table 1).

**Table 1.** The value of the *p* factor.

| Land Use Type | *p* Value | References |
|---|---|---|
| Construction land | 0 | Lu et al., 2013 [40]; Lin et al., 2020 [8] |
| Waterbody | 0 | Lu et al., 2013 [40]; Lin et al., 2020 [8] |
| Cropland | 0.4 | Xu et al., 2013 [41]; Chen and Zha, 2016 [42]; Lin et al., 2020 [8] |
| Grassland | 1 | Dai et al., 2013 [43]; Sun et al., 2014 [33]; Zhang et al., 2016 [44] |
| Forest | 1 | Xu et al., 2013 [41]; Dai et al., 2013 [43]; Sun et al., 2014 [33] |
| Marsh | 1 | Xu et al., 2013 [41]; Lin et al., 2020 [8] |
| Other land | 1 | Lu et al., 2013 [40]; Sun et al., 2014 [33]; Bamutaze et al., 2021 [14] |

*2.4. The Method for Analyzing the Impact of LC on Soil Erosion*

As the values of soil erosion are continuous, the impact of LC on soil erosion can be estimated by a multiple regression model using Stata 15.1. The multiple regression model used in this study can be written as:

$$y_i = \alpha + \beta LC_i + \gamma X_i + u_i \tag{2}$$

where *i* denotes different sampling sites, regardless of whether LC is carried out or not; the dependent variable $y_i$ denotes the annual soil losses; the independent variable LC represents whether LC is carried out; control variable $X_i$ captures other important factors in determining soil erosion intensity, and consist of types of soil, slope, rainfall, and vegetation cover [45]; $\alpha$, $\beta$ and $\gamma$ are the parameters that need to be estimated; and $\mu_i$ is the error term.

To estimate the heterogeneity effect of LC on soil erosion, the following multiple regression model was used in this study,

$$y_i = \alpha + \beta LC'_i + \gamma X_i + \varepsilon_i \tag{3}$$

where *i* denotes the different sampling sites where LC is carried out; the independent variable LC′ includes the variables of the type of LC project, the investment in the LC project, and the areas of newly increased cultivated land; $y_i$ and $X_i$ are the same as in Equation (2); $\alpha$, $\beta$ and $\gamma$ are the parameters that need to be estimated; and $\varepsilon_i$ is the error term. Definitions and descriptive statistics of all of the variables are presented in Table 2.

**Table 2.** Definitions and descriptive statistics of the variables.

| Variable | | Definition and Unit |
|---|---|---|
| **Dependent Variable** | **Erosion** | **Annual Soil Losses (t·ha$^{-1}$·yr$^{-1}$)** |
| Independent variables | LC | Whether LC is carried out (1 = yes; 0 = no) |
| | Type[①] | Types of LC (1 = farmland consolidation; 2 = land reclamation; 3 = land development) |
| | Invest | Investments of LC project per unit area (10,000 yuan/ha) |
| | Farmland | Areas of newly increased cultivated land (ha) |
| Control variables | Soil | Types of soil (1 = paddy soil; 2 = yellow-brown soil; 3 = limestone soil) |
| | Slope | Slope (°) |
| | Rainfall | Annual rainfall (mm) |
| | Cover | Vegetation cover (dimensionless) |

Note: ① Farmland consolidation works on existing cultivated land, land reclamation converts vacant/idle construction land and disaster-damaged land to cultivated land, and land development works on unused land to form newly cultivated land.

## 3. Results

### 3.1. Estimating the Soil Erosion

According to the RUSLE model, the rainfall erosivity factor (*R*), length–slope factor (*LS*), cover management factor (*C*), and practice factor (*P*) were used to calculate soil erosion. The distribution of the *R* factor, *LS* factor, *C* factor, and *P* factor is shown in Figure 3. From the *R* factor map (Figure 3a), the maximum value of the *R* factor is 7604.48 MJ·mm·ha$^{-1}$·hr$^{-1}$·yr$^{-1}$ and tends to decrease from the southeast to the northwest. The lowest R-value observed is 7233.52 MJ·mm·ha$^{-1}$·hr$^{-1}$·yr$^{-1}$. For the *LS* factor distribution (Figure 3b), the values range from 0 to 25.37. Most of the study area has a low *LS* factor value, and the lower value is located in most of the districts with a gentle slope, contributing to low–moderate soil loss. A high LS performs point-like distribution, which is more susceptible to soil erosion. Rich vegetation cover could effectively resist soil erosion. In the *C* factor map (Figure 3c), the value ranges from 0 to 1. Higher levels of *C* occur at water bodies and near human settlements with less vegetation cover, indicating high vulnerability to soil erosion. Lower levels of *C* occur in forests and concentrated areas of farmland, indicating less susceptibility to soil erosion. The *P* factor value reflects the impact of management practices on erosion. The *P* factor values of different land use types were obtained from previous studies, and the distribution is presented in Figure 3d.

The spatial pattern of the soil erosion in Lishui District is shown in Figure 4 through the RUSLE model, applying the rainfall erosivity factor, length–slope factor, cover management factor, soil erodibility factor, and practice factor. The soil erosion ranges from 0 to 385.77 t·ha$^{-1}$·yr$^{-1}$. As there are various RUSLE parameters, soil erosion has obvious spatial differences in distribution. Most of the study area has low levels of soil erosion values. Higher soil erosion values form a point-like distribution, especially in some areas with complex terrains. More soil erosion could occur in the central mountain area.

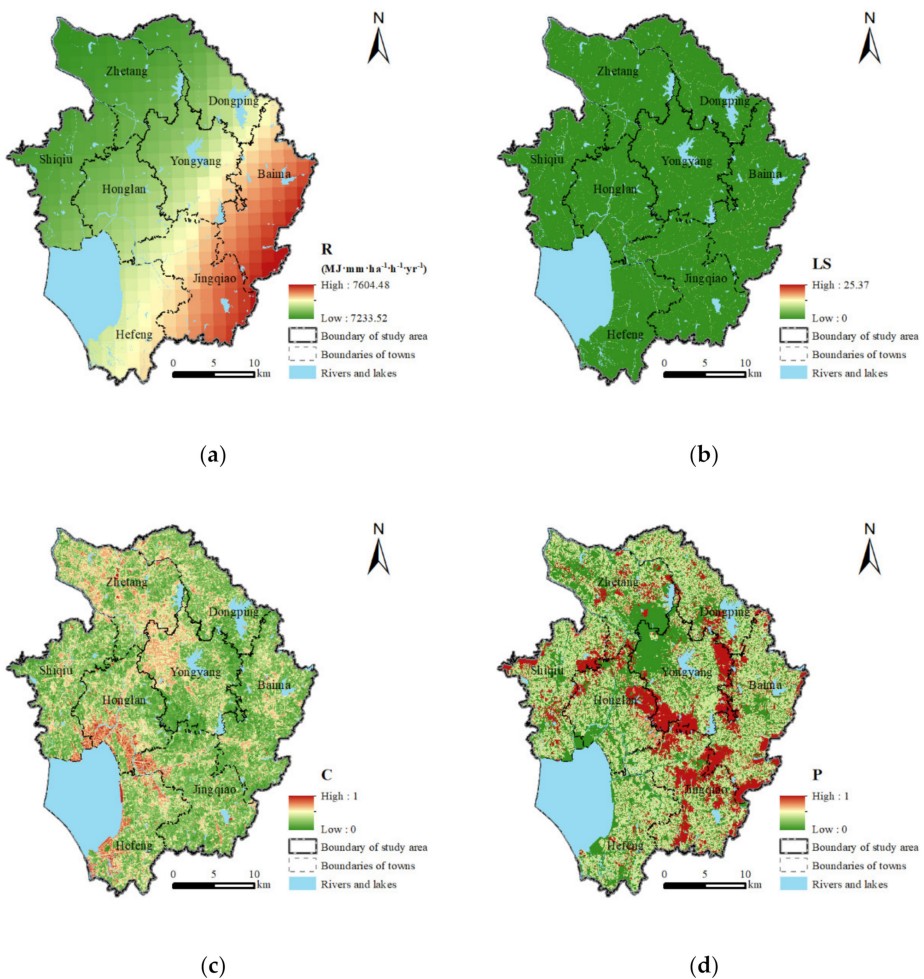

**Figure 3.** (**a**) Spatial distribution of the rainfall erosivity factor (*R*); (**b**) spatial distribution of the length–slope factor (*LS*); (**c**) spatial distribution of the cover management factor (*C*); (**d**) spatial distribution of the practice factor (*P*).

### 3.2. Estimating the Impact of LC on Soil Erosion

The results of the impact of LC on soil erosion are presented in Table 3. To test the robustness of the results, 300 points of agricultural land were randomly selected from the areas with and without LC, respectively; each was sampled twice and combined to form four groups of samples (Samplings 1–4). The coefficient of LC was $-0.304$ and statistically significant at the 5% level of significance, indicating that LC has a negative impact on annual soil losses (Model 1). This implies that LC has a positive influence on soil erosion reduction, where the amount of soil loss in plots with LC is lower than that in plots without LC. After adding the control variables, the coefficient of LC reached $-0.331$ and was statistically significant at a 5% level of significance, indicating that the addition of control variables did not affect the regression results, and the regression coefficient of LC remained negative (Model 2). Additionally, the results of Models 3–8 showed that the effect of LC on soil erosion was robust and did not change with the change in sampling, although the coefficient of LC in Samplings 3 and 4 failed to pass the significance test, it remained negative.

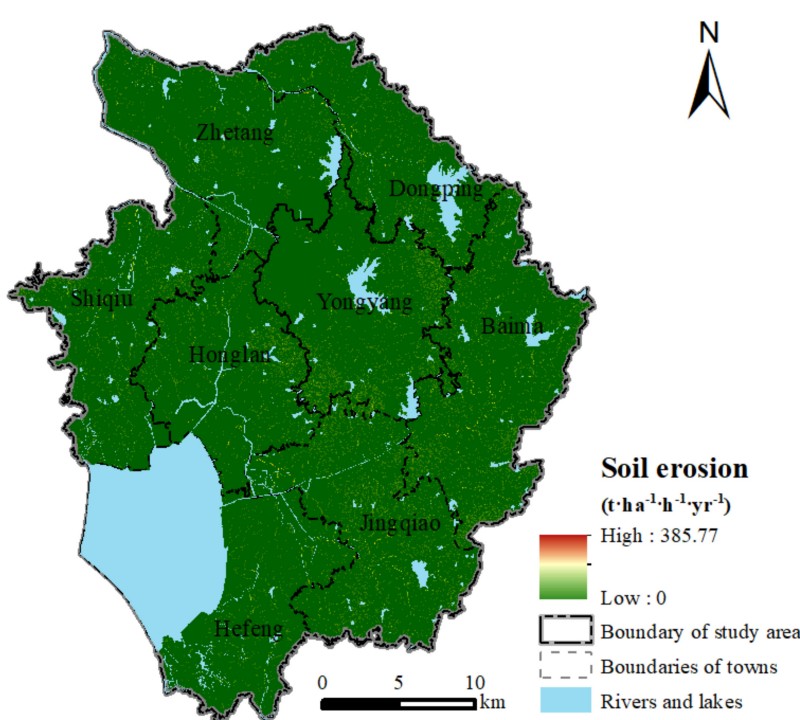

**Figure 4.** Soil erosion spatial distribution of Lishui District.

**Table 3.** Results of the impact of LC on soil erosion.

| Erosion | Sampling 1 | | Sampling 2 | | Sampling 3 | | Sampling 4 | |
|---|---|---|---|---|---|---|---|---|
| | Model 1 | Model 2 | Model 3 | Model 4 | Model 5 | Model 6 | Model 7 | Model 8 |
| LC | −0.304 ** (−1.98) | −0.331 ** (−2.13) | −0.388 ** (−1.99) | −0.393 ** (−1.98) | −0.164 (−0.98) | −0.225 (−1.30) | −0.248 (−1.20) | −0.319 (−1.51) |
| Paddy soil | | 0.033 (0.04) | | 0.104 (0.08) | | 0.316 (0.34) | | 0.499 (0.39) |
| Yellow-brown soil | | 0.404 (0.47) | | 0.195 (0.16) | | 0.413 (0.44) | | 0.265 (0.21) |
| Slope | | 0.024 (0.97) | | 0.005 (0.19) | | 0.037 (1.31) | | 0.011 (0.35) |
| Rainfall | | 0.004 (0.43) | | 0.021 * (1.96) | | 0.018 * (1.95) | | 0.034 *** (3.12) |
| Cover | | 0.422 (0.68) | | −0.345 (−0.43) | | 0.604 (0.96) | | 0.039 (0.05) |
| Constant | 0.717 *** (6.61) | −4.519 (−0.41) | 0.802 *** (5.80) | −25.602 * (−1.87) | 0.717 *** (6.08) | −22.670 * (−1.95) | 0.802 *** (5.50) | −43.336 *** (−3.07) |
| Observations | 600 | 600 | 600 | 600 | 600 | 600 | 600 | 600 |

Notes: *t*-statistics in parentheses; *** $p < 0.01$, ** $p < 0.05$, * $p < 0.1$.

The results of the heterogeneity effects of LC on soil erosion are presented in Table 4, where only the types of LC projects show a significant impact on soil erosion. The coefficient of farmland consolidation was 0.561 and statistically significant at the 5% level of significance, indicating that, compared to land development, farmland consolidation leads to more severe soil losses (Model 11). Although the coefficient of farmland consolidation in Model 9 did not pass the significance test, it remained positive. After adding the control variables, the coefficient of farmland consolidation reached 0.578 and is statistically significant at the 5% level of significance, indicating that the addition of control variables did not affect the regression results, and the regression coefficient of farmland consolidation remained positive (Model 12).

**Table 4.** Results of the heterogeneity effect of LC on soil erosion.

| Erosion | Samplings 1 and 2 | | Samplings 3 and 4 | |
|---|---|---|---|---|
| | Model 9 | Model 10 | Model 11 | Model 12 |
| Farmland consolidation | 0.127 (0.57) | 0.112 (0.50) | 0.561 ** (2.03) | 0.578 ** (2.09) |
| Land reclamation | 0.102 (0.40) | 0.066 (0.26) | 0.091 (0.26) | 0.160 (0.45) |
| Invest | −0.004 (−0.35) | −0.001 (−0.10) | −0.004 (−0.23) | −0.001 (−0.07) |
| Farmland | 0.007 (0.40) | 0.009 (0.48) | −0.004 (−0.21) | −0.006 (−0.33) |
| Paddy soil | | −0.165 (−0.25) | | 0.271 (0.31) |
| Yellow-brown soil | | 0.259 (0.39) | | 0.087 (0.10) |
| Slope | | 0.002 (0.07) | | 0.023 (0.68) |
| Rainfall | | 0.007 (0.84) | | 0.035 *** (3.35) |
| Cover | | 0.245 (0.35) | | 0.790 (0.99) |
| Constant | 0.333 (1.39) | −8.940 (−0.82) | 0.322 (1.07) | −46.081 *** (−3.35) |
| Observations | 300 | 300 | 300 | 300 |

Notes: *t*-statistics in parentheses; *** $p < 0.01$, ** $p < 0.05$.

## 4. Discussion

### 4.1. Positive Effects of LC on Soil Erosion Reduction

In this study, compared to plots without LC, plots with LC showed lower soil erosion (Figure 5a). The positive effects of LC on soil erosion reduction could be explained from the perspectives of vegetation cover and engineering projects.

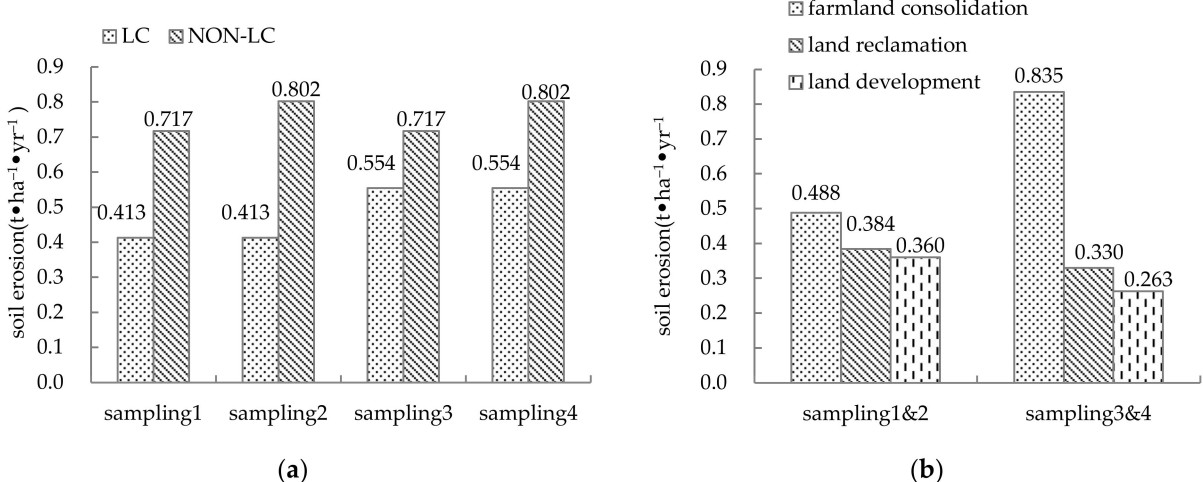

**Figure 5.** (**a**) Soil erosion between plots with and without LC and (**b**) soil erosion between different types of LC.

The construction of protection forests, farmland shelterbelts, or vegetation protection systems is one of the basic requirements for LC [17,46,47], and the retention and increase of vegetative cover of soil is a vital factor for maintaining soil stability [48] and for reducing

the dynamics of runoff [49], thus it is conducive to soil and water conservation. During the process of LC in Lishui District, vegetation cover, such as trees, shrubs, and grasses, are planted to protect the surface of slopes and to reduce soil erosion.

The construction of differentiated land engineering is one of the major types of LC that is widely carried out in China. First, projects of land leveling can reduce the altitude difference between plots and can make land flatter, thus decreasing the flow rate and weakening water erosion. Second, with LC in sloped areas, especially areas with high hypsography, measures of terraces or other ecological slope protection projects are taken to improve water conservation, avoid land collapse, and control soil erosion [47,49,50]. Third, gully consolidation projects create more farmland in gullies and reduce land reclamation on slopes, which is helpful for soil conservation on slopes [51,52], and filling gullies for farmland could indeed reduce the soil erosion at the bottom of such gullies [53]. Fourth, soil reconstruction projects can improve soil particle composition and profile structure [54], and a favorable soil structure contributes to water and nutrient retention as well as a decrease in erodibility [55]. Last but not least, drainage engineering carried out in appropriate locations is conducive to alleviating problems with ponding caused by short-term intense rainfall under heavy rain events [27,49]. During the process of LC in Lishui District, different kinds of farmland consolidation, land reclamation, and land development were carried out. Soil erosion in the plots with different types of LC (farmland consolidation, land reclamation, or land development) was lower than that in the plots without LC, except for farmland consolidation in Sampling 3 and 4 (Figure 5b).

### 4.2. Potential Risks Brought by LC to Soil Erosion Reduction

LC in China is used to reduce land fragmentation, increase cultivated land, improve production capacity, and strengthen intensive land use [21,56,57]. Due to these goals, there may be several potential risks that hinder the soil erosion reduction during or after LC.

First, different tillage systems have different impacts on soil compaction and soil erosion, while runoff and erosion reduce under decreasing tillage intensity [58,59]. Compared to conventional tillage systems, no-till and conservation tillage could decrease soil erosion on sloping agricultural land [60–62]. As a reverse process of intensive land use, the marginalization and abandonment of land could increase vegetation cover and reduce soil erosion [48,63]. The results of Han et al. (2020) also showed that cultivated land experienced stronger erosion than abandoned land or forest–grass land [49]. However, what needs to be noted is that abandoned land increases erosion when the soil is left bare [48,64]. Land development may become beneficial if the abandoned/unused land in one area is bare and with the prerequisite of guarding against desertification and soil erosion [65]. It may be one of the possible reasons for farmland consolidation in some areas leading to more severe soil erosion, while land development leads to slighter soil erosion (Models 11–12; Figure 5b).

Second, LC may lead to increases in the use of fertilizers, pesticides, and plastics when newly increased cultivated land is of poor quality [66]. On the one hand, the misuse of fertilizers, pesticides, and especially herbicides may induce environmental problems and soil erosion [48,61]. The study by Keesstra et al. (2016) also showed that the highest runoff and soil erosion was both identified in the herbicide-treated plots (compared to tillage plots and covered plots) [64]. On the other hand, the plastic film mulching used in agricultural land could intensify soil erosion and embankment collapse due to the rapidly formed concentrated flow under heavy rain, especially with inappropriate drainage systems [49].

### 4.3. Strategies for LC in the Future

LC has turned out to be more important in developing countries, including China, which has the characteristics of soil degradation, and with the continuous and increasing attention to ecological environment issues, ecological environment protection has become an indispensable part of LC. Strategies for LC in the future should fully take the economy,

ecology, and sustainability into consideration, and the goals of improving the quantity, quality, and ecological function of land should be clear and definite.

On the one hand, the proper use of cultivated land after LC should be guaranteed. The cultivated land is mainly used for grain production, and the aim of LC is to improve the grain production infrastructure and increase productivity. The phenomenon of using cultivated land to plant fruit trees, tea plants, or nursery stocks is contrary to the original intention of LC, which needs to be eliminated. Moreover, suitable tillage methods should be adopted in cultivated land after LC. Conservation tillage is recommended for sloping cultivated land to reduce soil erosion. Additionally, LC is not a temporary project, and it is necessary to achieve the continuous management and protection of land after LC.

On the other hand, the concept of "mountains, waters, forests, farmlands, lakes, and grasslands are part of a community of life" and the laws of "natural recovery and ecological priority" [67] should be complied with during the process of LC. In the current practice of LC, more attention is still paid to eliminating or weakening restrictive factors in development (low efficiency of resource utilization, restoration of ecological environment damage, etc.). LC in the future should attach importance to the combination of comprehensive regulation of fields, water, roads, forests, and villages and the construction of soil and water conservation, wind, and sand fixation in important ecological functional areas.

Additionally, the impact of LC shows obvious regional differences and measure differences. Differentiated measures should be taken to improve the quality of cultivated land, rationally exploit abandoned land, rationally construct engineer land, and realize the remediation of land degradation and restoration of land ecology.

## 5. Conclusions

The hilly areas of China experience soil erosion, which harms production, livelihoods, and ecology. Taking Lishui District, Nanjing City, as the study area, this study quantitatively evaluated the soil losses in the study area. The soil erosion ranged from 0 to 385.77 t·ha$^{-1}$·yr$^{-1}$ with spatial heterogeneity due to the topography, land cover, and vegetation cover.

The hilly areas in China are also typical LC regions, and LC is one of the important human factors affecting soil erosion. This study further analyzed the impact and its heterogeneity of LC on soil erosion. LC has reduced soil erosion in the study area, where the amount of soil loss in plots with LC is lower than that in plots without LC. Moreover, farmland consolidation could lead to more serious soil erosion compared to land development. Overall, LC is conducive to reducing soil erosion due to the construction of protection forests or farmland shelterbelts, and differentiated land engineering, which could increase vegetative cover, decrease flow rate and alleviate ponding problems, maintain soil stability, and improve soil structure, or protect slopes and avoid collapse. However, there may be potential risks brought by LC to soil erosion reduction, such as harm vegetation from intensive land use and herbicide or plastic misuse, runoff, and erosion. Therefore, strategies for LC in the future should harmonize the promotion of LC and the protection of the ecological environment.

**Author Contributions:** Y.Z. and M.X. designed the paper; C.X. collected the data and contributed to the RUSLE model; Y.Z. and C.X. wrote the paper; M.X. revised the paper. All authors have read and agreed to the published version of the manuscript.

**Funding:** This study was supported by the National Natural Science Foundation of China (NSFC) (Grant No. 72003090).

**Institutional Review Board Statement:** Not applicable.

**Informed Consent Statement:** Not applicable.

**Data Availability Statement:** The DEM data presented in this study are openly available in Geospatial Data Cloud, which is available online (http://www.gscloud.cn/, accessed on 12 January 2021). The rainfall data presented in this study are openly available in Meteorological Data Service Center,

which is available online (http://data.cma.cn/, accessed on 12 January 2021). The rest of the data are available on request from the corresponding author.

**Conflicts of Interest:** The authors declare no conflict of interest.

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
