# Peer review of "Can Land Consolidation Reduce the Soil Erosion of Agricultural Land in Hilly Areas? Evidence from Lishui District, Nanjing City"

_land, doi:10.3390/land10050502_

Round 1

Reviewer 1 Report

Interesting study about the impact of land consolidation on soil erosion. The text is well structured with some minor language issues.

The introductory section can be improved. Although the methodology for estimating soil erosion (RUSLE) is introduced, however the methodology use for analyzing the data (Tobit model) is not discussed in the introduction (some background and motivation for using it). 

Materials and methods: Concerning the study site area, a summary about the land use types as well as the relevant NDVI estimated by the satellite image analysis is not provided. This information is highly relevant to RUSLE implementation. The motivation for the use of Tobit model as well as the how it meets the studies needs, requires some further explanation (i.e. why OLS was not selected)? Moreover, with the concept of left censoring it is implied that a data point can be identified below a certain limit value. Given that the limit value for soil erosion is settled to 0, I understand that the interpretation of the left censoring implies that may appear a soil erosion value below zero. It seems that such an interpretation contradicts the physical notion of soil erosion. 

Results: The section interpreting tables 3 and 4 (paragraph3.2) needs further elaboration (i.e. the interpretation of the different models, the relation significance between the variables and soil erosion etc) 

Discussion: The discussion section seems to have no tight connection to the rest of the study. It talks in detail about the various kinds of consolidation measures, however it does not really connect them to the current work, i.e average soil erosion identified in areas with specific type of consolidation measures, neither is there some parallelism with other relevant studies referring to specific consolidation measures. The same applies also for the identified relations of soil erosion with the studied variables. 

Reviewer 2 Report

See attached file

Round 2

Reviewer 1 Report

The work was significantly improved.

Minor issues that require authors' attention are:

line 159: Eq. 6 - the number of the equation can not be correct. Please correct.

Lines 196 & 210: no past tense is needed, I would suggest "...soil erosion is presented in..."

Line 212: redundant "s" in the sentence "...newly increased cultivated land on s soil erosion were investigated"

Line 231: minor restructuring of the sentence is needed - i.e. in this study, the plots with LC have lower soil erosion compared to the plots without LC.

Lines 255 - 256 (and Figure 5): Graphs 5a and 5b are missing the units of the measured variable (soil erosion). Which is the value of soil erosion without land consolidation? Assuming that the average soil erosion value in non-consolidation is around 0.8 (as may derive from fig 5a), farmland consolidation for sampling 3, and 4 (fig. 5b) exceeds this value (0.835). Thus the statement in lines 255 - 256 seems not to hold. Please clarify this point.

Line 264: maybe the authors could replace the ".. will reduce..." with  "...are reducing..." since it is a general case

Lines 292 - 294: the sentence in these lines may need some reshaping, for more clear meaning. 

Congratulations for your nice work.

Reviewer 2 Report

The authors have followed most of the recommendations given. Therefore, as a consequence of the changes and corrections incorporated in this new version, the scientific quality of the manuscript has considerably improved. However, the title still does not seem appropriate and, in my opinion, before being published the manuscript needs a grammar revision.

Author Response

This manuscript is a resubmission of an earlier submission. The following is a list of the peer review reports and author responses from that submission.